# The Clinical Impact of the Pulmonary Embolism Severity Index on the Length of Hospital Stay of Patients with Pulmonary Embolism: A Randomized Controlled Trial

**DOI:** 10.3390/diagnostics14070776

**Published:** 2024-04-07

**Authors:** Marco Paolo Donadini, Nicola Mumoli, Patrizia Fenu, Fulvio Pomero, Roberta Re, Gerardo Palmiero, Laura Spadafora, Valeria Mazzi, Alessandra Grittini, Lorenza Bertù, Drahomir Aujesky, Francesco Dentali, Walter Ageno, Alessandro Squizzato

**Affiliations:** 1Thrombosis and Haemostasis Center, Ospedale di Circolo, ASST Sette Laghi, 21100 Varese, Italy; marcopaolo.donadini@uninsubria.it (M.P.D.); francesco.dentali@uninsubria.it (F.D.); walter.ageno@uninsubria.it (W.A.); 2Research Center on Thromboembolic Diseases and Antithrombotic Therapies, University of Insubria, 21100 Varese and 22100 Como, Italy; lorenza.bertu@uninsubria.it (L.B.); alessandro.squizzato@uninsubria.it (A.S.); 3Department of Internal Medicine, Magenta Hospital, 20013 Magenta, Italy; alessandra.grittini@asst-ovestmi.it; 4Presidio Ospedaliero di Livorno, Azienda USL Toscana Nord Ovest, 57124 Livorno, Italy; mazzivaleria@gmail.com; 5Presidio Ospedaliero di Cecina, Azienda USL Toscana Nord Ovest, 57023 Cecina, Italy; patrizia.fenu@uslnordovest.toscana.it; 6Internal Medicine Unit, Michele e Pietro Ferrero Hospital, 12060 Verduno, Italy; fulviopomero@yahoo.it (F.P.); lauretta.spadafora@gmail.com (L.S.); 7Medicina Interna, Ospedale S. Andrea, ASL Vercelli, 13100 Vercelli, Italy; roberta.re@aslvc.piemonte.it; 8Ospedale Versilia, Azienda USL Toscana Nord Ovest, 55049 Viareggio, Italy; gerardo2002@libero.it; 9Department of General Internal Medicine, Bern University Hospital, University of Bern, 3010 Bern, Switzerland; drahomir.aujesky@insel.ch; 10Internal Medicine Unit, ‘Sant’Anna’ Hospital, ASST Lariana, 22042 San Fermo della Battagli, Italy

**Keywords:** pulmonary embolism, Pulmonary Embolism Severity Index, length of hospital stay

## Abstract

Background: The Pulmonary Embolism Severity Index (PESI) is an extensively validated prognostic score, but impact analyses of the PESI on management strategies, outcomes and health care costs are lacking. Our aim was to assess whether the adoption of the PESI for patients admitted to an internal medicine ward has the potential to safely reduce the length of hospital stay (LOS). Methods: We carried out a multicenter randomized controlled trial, enrolling consecutive adult outpatients diagnosed with acute PE and admitted to an internal medicine ward. Within 48 h after diagnosis, the treating physicians were randomized, for every patient, to calculate and report the PESI in the clinical record form on top of the standard of care (experimental arm) or to continue routine clinical practice (standard of care). The ClinicalTrials.gov identifier is NCT03002467. Results: This study was prematurely stopped due to slow recruitment. A total of 118 patients were enrolled at six internal medicine units from 2016 to 2019. The treating physicians were randomized to the use of the PESI for 59 patients or to the standard of care for 59 patients. No difference in the median LOS was found between the experimental arm (8, IQR 6–12) and the standard-of-care arm (8, IQR 6–12) (*p* = 0.63). A pre-specified secondary analysis showed that the LOS was significantly shorter among the patients who were treated with DOACs (median of 8 days, IQR 5–11) compared to VKAs or heparin (median of 9 days, IQR 7–12) (*p* = 0.04). Conclusions: The formal calculation of the PESI in the patients already admitted to internal medicine units did not impact the length of hospital stay.

## 1. Introduction

Pulmonary embolism (PE) is a common cardiovascular disease with an estimated incidence of ~1 out of 1000 persons per year [1,2]. PE is associated with a wide prognostic spectrum, ranging from the prompt and complete resolution of symptoms after few hours of treatment to sudden death [3,4]. Patients with PE are commonly admitted to the hospital for their initial treatment, though some of them may be suitable for a short hospital stay or a complete home treatment [5,6,7,8]. Indeed, in recent years, research has focused on stratifying the risk of adverse outcomes associated with PE to tailor treatment and management strategies. Although some prognostic scores have been adequately derived and validated, especially the PESI score (Pulmonary Embolism Severity Index) [7,9,10,11], there is no evidence that the use of these scores changes physicians’ behaviors and improves patient outcomes and/or reduces health care costs [12,13]. The PESI calculation is currently recommended by clinical practice guidelines as a tool to identify patients with PE who are at low risk of short-term adverse outcomes and may be discharged early [7]. Of note, many additional factors may have an impact on the possibility to safely allow for the early discharge of patients with low-risk acute PE, including the choice of anticoagulant drugs, adequate family and/or home care support, co-morbidities, the need for oxygen supply, and pain control. Indeed, the duration of hospitalization for PE still remains long in many clinical contexts [8,14,15,16,17].

Parenteral anticoagulants and direct oral anticoagulants (DOACs) are the current options for the acute treatment of non-high-risk PE [18,19,20,21]. Parenteral drugs (i.e., unfractionated heparin (UFH), low-molecular-weight heparin (LMWH) and fondaparinux) are efficacious but not optimal for home treatment [6,7]. DOACs have simplified the management of PE thanks to their pharmacologic properties (rapid onset of action, short half-life, and predictable anticoagulant effect) compared to vitamin K antagonists (VKAs).

Our hypothesis was that the use of a validated clinical prediction model to stratify acute PE prognosis would have an impact on the attitudes of clinicians and on the management of patients with PE who were already admitted to the hospital. We postulated that physicians would be able to optimize the duration of hospital stay for PE by shortening it, thus potentially also reducing hospital-related complications and costs without increasing the risk of PE-related adverse outcomes. Furthermore, we postulated that DOACs, as opposed to VKA treatment, may simplify, and thus promote, home treatment for the acute phase of PE.

## 2. Methods

The Impact Analysis of Prognostic Stratification for Pulmonary Embolism (iAPP) study is a randomized, parallel-group, open-label trial that was conducted from 2016 to 2019 at internal medicine units of six Italian hospitals from different provinces (Livorno, Viareggio, Cecina, Novara, Cuneo, and Magenta) that were already part of a collaborative study group evaluating the length of hospital stay for patients with PE [14].

The study protocol was approved by the Ethics Committee of Insubria, Varese, Italy and by all of the participating centers. This study was conducted following the Good Clinical Practice rules and in agreement with the Declaration of Helsinki.

### 2.1. Participants

Consecutive adult outpatients with an objectively confirmed diagnosis of suspected or unsuspected acute PE at the emergency department (ED) and subsequently admitted to one of the participating internal medicine units were eligible and enrolled after providing written informed consent. No exclusion criteria were applied.

Suspected PE was defined as a diagnosis of PE confirmed by an imaging test (i.e., computed tomographic pulmonary angiography [CTPA], pulmonary angiography, or V/Q lung scan) prescribed by a physician who had a clinical suspicion of PE. Unsuspected PE was defined as a diagnosis of PE that was made incidentally by an imaging test performed for other clinical indications (e.g., cancer staging or follow-up; investigation for chest diseases other than PE).

An objective diagnosis of acute PE was defined as the presence of at least one intra-luminal filling defect of pulmonary arteries at CTPA or pulmonary angiography, a high-probability ventilation/perfusion (V/Q) lung scan (or perfusion lung scan with negative chest X-ray), or an intermediate-probability V/Q or perfusion lung scan with proximal deep-vein thrombosis (DVT) documented by ultrasonography.

### 2.2. Interventions

Within 48 h of an acute PE diagnosis, a local investigator (treating physician) was centrally randomized for every included patient to the experimental approach, i.e., formal PESI calculation and documentation of the PESI in the clinical record form on top of routine clinical practice, or to the standard of care, i.e., no routine calculation of the PESI. A pre-printed form for annotating the PESI score, including the corresponding short-term mortality according to the PESI class, was filled out and added to the clinical record form of patients randomized to the experimental arm (available upon request).

### 2.3. Outcomes

The primary efficacy outcome was the median length of hospital stay (LOS).

Secondary efficacy outcomes included the proportions of patients undergoing a short hospital stay (i.e., <48 h in hospital), the proportion of post-discharge outpatients visiting the emergency department, the hospital re-admission rate within 90 days, and quality of life (5-point Likert scale questionnaire)

Other outcomes were represented by in-hospital and 90-day overall mortality, recurrent PE and/or DVT, major bleeding (according to the International Society on Thrombosis and Haemostasis (ISTH) criteria [22]), and other anticoagulation-related complications (hematoma/infection at heparin or fondaparinux injection sites or heparin-induced thrombocytopenia).

Additional hospitalization-related outcomes were recorded, including hospital-acquired infections (pneumonia; urinary tract infection; or other), iatrogenic complications, immobilization syndrome, and pressure sores.

### 2.4. Sample Size

In a previously published observational study conducted by the same study group [14], the median hospital stay for PE in internal medicine units was 12 days (interquartile range [IQR] of 9-17), which was concordant with the administrative data from the Lombardia Region of the mean LOS of 11.5 days [15]. Based on those data, we hypothesized that the mean LOS would be reduced by at least 15% in all patients with PE managed with the formal calculation of the PESI score and by 5% in the standard-of-care arm (because of increased knowledge of PE prognostic stratification in recent years). Therefore, with an α error of 0.05 and a statistical power (1-β error) of 80%, 200 patients in each group (a total of 400 patients) were estimated to be necessary to find a statistically significant difference (*p* < 0.05) between the mean LOS of the two experimental arms. As the variable LOS has an expected non-normal distribution and needs to be expressed and reported as a median, 10% extra patients were needed to reach a statistically significant difference with the previous statistical assumptions. The final total sample size was therefore estimated to be 440 patients (220 patients for each group).

### 2.5. Randomization

Randomization was performed centrally with a 1:1 ratio following a computer-generated list of randomizations and was stratified by the previously declared anticoagulant treatment choice of the local investigator (i.e., LMWH +/− VKA (VKA group) vs. DOACs as single drug or with lead-in heparin (DOACs group) in order to prevent the treatment choice from having any influence on the final results.

The allocation was concealed to the local investigators. The list of randomizations was maintained only by the study coordinator (AS), who assigned the treating physician to the management arm according to the randomization sequence, after being notified by any local investigator of a new patient’s enrolment and anticoagulant treatment choice.

### 2.6. Statistical Analysis

The categorical variables measured are expressed as numbers and percentages. Continuous variables are reported as means (standard deviation) or medians (interquartile range [IQR]) depending on the normal distribution of the data.

The primary outcome was analyzed across the study groups by means of the Mann–Whitney U test.

Additional analyses were performed using the chi-square test or unpaired *t*-test, as appropriate. As a prespecified secondary analysis, the LOS was also compared between the DOACs group versus the VKA group.

A statistical analysis was performed by using the IBM SPSS Statistics software, version 27 (SPSS, Inc., IBM corporation, U.S., Armonk, NY, USA).

## 3. Results

This study was prematurely stopped after reaching 27% of the planned sample size due to a slow recruitment rate, which made it unfeasible to reach the originally planned number of patients.

From July 2016 to October 2019, 125 patients who were consecutively admitted to the six participating internal medicine units within 48 h of a PE diagnosis were enrolled in the trial. For 7 patients, the local investigators did not perform study procedures after randomization, thus leaving 118 allocated patients. The local investigators were randomized to formally calculate and report the PESI score on top of the standard of care for 59 patients or to use the standard of care alone for 59 patients. No patient was excluded after allocation, and all data were available for the primary outcome (Figure 1).

Overall, the study population included 62 males (52.5%) and 56 females (47.5%), with a mean age of 75.6 years (SD 12.8). PE was incidentally diagnosed in 20 patients (16.9%) and was provoked by at least one major risk factor in 40 patients (33.9%). Concomitant DVT was diagnosed in 65 patients (55.1%). The clinical presentation of PE was characterized by sustained hypotension (i.e., high-risk PE) in three patients (2.5%). The baseline characteristics of the study population according to the assigned management arm are presented in Table 1.

The median LOS was 8 days (IQR 6–12). No difference was found between the experimental arm (median of 8 days, IQR 6–12) and the standard-of-care arm (median of 8 days, IQR 6–12) (*p* = 0.63). No patient was discharged within 48 h of PE diagnosis.

The mortality rate at 90 days was 6.8% (8 patients) and did not differ between the two arms (*p* = 0.48). Recurrent VTE occurred in two patients within the standard-of-care arm (3.4%).

Two patients experienced major bleeding, both in the standard-of-care arm (3.4%).

All secondary efficacy outcomes and safety outcomes are presented in Table 2 according to the randomization arm.

Data on the discharge destinations and on the clinical and family/social determinants of the LOS according to the randomization arm are presented in Table 3 and Table 4.

There were no significant differences between the randomization groups on the quality of life items (Table 5).

### Secondary Analysis

DOACs were used in 71 patients (with or without lead-in heparin, according to indication), whereas parenteral anticoagulants alone or followed by VKA were used in 47 patients. Patients treated with DOACs, compared to those treated with parenteral anticoagulation alone or followed by VKA, were significantly younger (72.4 vs. 80.5 years; *p* 0.0006) and had a lower prevalence of active cancer (11.3% vs. 31.9%; *p* 0.001). Moreover, a significantly higher proportion of patients were classified as low-risk PESI classes among those treated with DOACs (28 patients, 39.4%) compared to those treated with parenteral anticoagulation alone or followed by VKA (7 patients, 14.9%).

The LOS was significantly shorter among patients treated with DOACs (median of 8 days, IQR 5–11) compared to those treated with parenteral anticoagulation (median of 9 days, IQR 7–12). Patients that were treated with DOACs also had a significantly lower incidence of pressure sores and in-hospital and 90-day mortality rates (Table 6).

## 4. Discussion

Our study suggests that the mere calculation and documentation of the PESI in patients with PE who were already admitted to internal medicine units may not have a clinically relevant impact on the duration of hospital stay. However, since this study was prematurely interrupted and did not reach the planned sample size, the results do not allow for any firm conclusion to be drawn.

Since 2008, the European Society of Cardiology (ESC) [23] has proposed a stepwise risk stratification approach to optimize the management of patients with PE, using a combination of clinical findings, imaging, and biochemical markers to distinguish between patients with high, intermediate, and low risks of an early adverse outcome. One of the most challenging tasks is to identify, within the large group of normotensive and apparently stable patients, those whose risk is ‘sufficiently low’ to permit early discharge and outpatient treatment [3,7,24]. Such an approach may minimize early complications related to hospitalization and may have an impact on health care costs as well as on patient satisfaction and quality of life. Clinical decision rules (CDRs) are the best available tools to combine clinical findings, and among them, the PESI and simplified PESI are recommended by current guidelines [7,25,26].

However, the PESI is used in clinical practice and recommended by current guidelines without any solid/definitive evidence that any CDR may assist clinicians in determining the best treatment and the appropriate setting for the initial therapy, except for an RCT in which e-health care record-based risk stratification of the PESI (plus teaching) has been shown to reduce hospital admissions [12,27]. Three steps are involved in the development and testing of a new CDR [12,13]. The first stage is derivation, where the independent and combined effects of explanatory variables such as symptoms, signs, and/or investigations are established. The second stage is validation, where the final derived CDR is evaluated first in different clinical settings. The final stage of evaluation is to test the impact of using the CDR in clinical practice, ideally in a randomized controlled trial (RCT), for relevant clinical outcomes. To the best of our knowledge, this is the first RCT that explored the clinical impact of PESI itself on inpatients with PE [3,6]. A recently published study used a combined strategy involving risk stratification by using the PESI followed by predefined criteria for mobilization and discharge, which was effective in reducing the LOS [28]. Indeed, even if our study was stopped prematurely, the results do not suggest that the PESI has any clinically relevant impact on the LOS. The PESI was primarily developed to avoid hospital admission for outpatients with PE, and not to decrease the LOS of patients with PE in a medical ward [9]. However, the lack of effect may be differently explained. Prognosis is only a medical description of a patient’s condition; indeed, discharge is not only based on the predictive risk of complications, but it is also a multiparametric choice based on social, family, and psychological factors. The lack of a caregiver at home, difficulties in performing imaging and lab tests outside the hospital, and a fear of being treated without nursing and/or medical assistance may be major determinants of the length of hospital stay. Another important factor is patients’ preferences; outpatient care or early discharge for a serious condition such as PE may be socially inacceptable to many patients or physicians. Finally, the question remains whether mere calculation and documentation, without any further educational support, is enough to change practice.

The difference in the LOS between VKA and DOAC in our study and in the literature seems to support this hypothesis [29]. Indeed, for many patients, the use of LMWH and fondaparinux requires daily nurse assistance for subcutaneous injections, and VKA treatment management may need more time for patients to be settled out of hospital compared to DOACs, in addition to the potential for the treating physician to wait for at least one therapeutic INR before discharging patients with acute PE.

Worldwide, several papers have reported a median length of hospital stay of 5 days or less for PE. In our population, the median length of hospital stay was approximately 8 days. It may be hypothesized that a major determinant of the length of hospital stay could be the organization of the health care system, as in Italy, the length of hospital stay has been similar in the past 10 years [15,16], as confirmed by similar LOSs in other European countries, such as France (11.6 days) or Spain (6.8 days) [29].

Our study has several limitations. First, this study was prematurely interrupted due to a slow recruitment rate; therefore, by definition, this study is underpowered to detect any difference among groups. Second, the main inclusion criterion was the admission to an internal medicine unit. In all recruiting centers, patients with PE are also admitted in other units, such as cardiology, pneumology, or intensive care units. Therefore, the results of our study may only be applied to these subgroups of admitted patients with PE, and whether this is also the case in patients hospitalized in cardiology or pneumology units, where patients may be more selected, younger, and less multimorbid, is still unknown. In addition, the PESI was primarily developed to avoid hospital admission for outpatients with PE and not to decrease the LOS of patients admitted to the medical ward. Third, the treating physicians were centrally randomized for every patient; some authors suggest that cluster randomization is the most appropriate study design to test the impact of a CPR to avoid the risk of contamination. Indeed, taking part in an interventional study improves the knowledge of the participating investigators on the topic irrespective of the study design and may potentially bias the results by reducing the difference between the intervention and control groups, i.e., physicians randomized to usual care might also calculate/use the PESI to determine the LOS. Moreover, contamination might also play a role, as physicians working at the same ward/hospital might also become contaminated in terms of their practices if they know what the study is about. Therefore, we included this consideration into the sample size calculation, as we anticipated that the LOS would have been reduced by 5% in the standard-of-care arm (compared to 15% in the interventional group); however, this estimate may not be accurate. Fourth, DOACs may facilitate outpatient care. However, the patients were not randomized based on the treatment received, and it is possible that the patients who were perceived to be at a low risk were more likely to be discharged early and to receive DOACS. So, whether the shorter LOS in the patients treated with DOACs is a consequence of the DOACs or confounded by less severe PE still needs to be determined. Indeed, there seemed to be fewer severe PE cases (66% vs. 75%) and more patients with incidental PE (24% vs. 10%) in the PESI group compared to the standard of care group, respectively (see Table 1)

In conclusion, the knowledge of the PESI in patients with PE who were hospitalized in internal medicine units did not appear to reduce the duration of hospital stay in our study. Future studies should assess the role of the PESI in other inpatient settings or explore its use in combination with other major potential determinants of hospital stay.

## Figures and Tables

**Figure 1 diagnostics-14-00776-f001:**
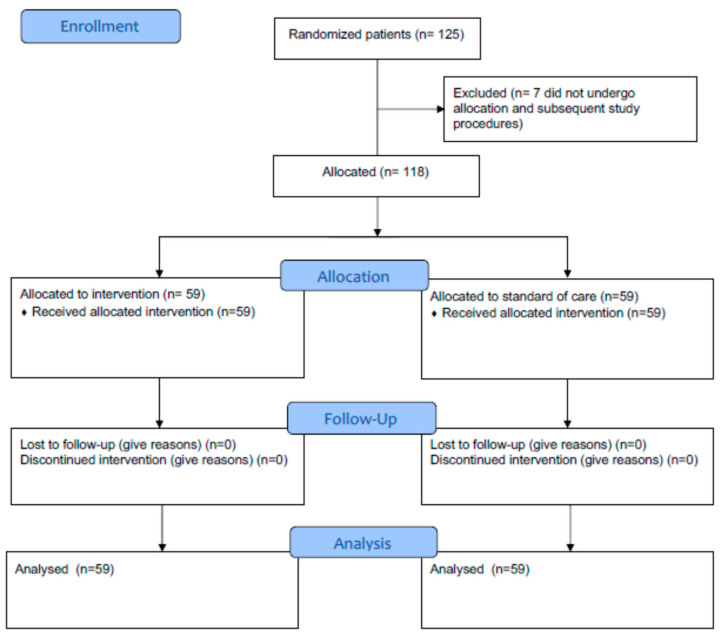
iAPP flow diagram.

**Table 1 diagnostics-14-00776-t001:** Baseline characteristics: DVT, deep vein thrombosis; PE, pulmonary embolism; DOAC, direct oral anticoagulant; RVD, right ventricular dysfunction; SBP, systolic blood pressure; TTE, transthoracic echocardiography; n, number.

Patients, n	PESI (n = 59)	Routine Practice (n = 59)	*p*
Age, ys—mean (SD)	75.6 (13.1)	75.7 (12.7)	0.966
Sex, female—n (%)	31 (52.5)	25 (42.4)	0.356
Weight—mean (SD)	75.4 (15.5)	76.3 (9.8)	0.706
Unprovoked PE—n (%)	38 (64.4)	40 (69.0)	0.845
Active cancer—n (%)	12 (20.3)	11 (18.6)	0.816
Concomitant DVT—n (%)	35 (59.3)	30 (50.9)	0.459
Incidental PE—n (%)	14 (23.7)	6 (10.2)	0.085
SBP < 90 mmHg	2 (3.4)	1 (1.7)	0.768
RVD * on TTE—n (%)	10 (21.3)	16 (33.3)	0.266
RVD on lab biomarkers	13 (48.2)	14 (37.8)	0.826
PESI—mean (SD)	115.1 (49.9)	115.4 (44.3) §	0.972
PESI class III, IV, V—n (%)	39 (66.1)	44 (74.6) §	0.420
DOAC treatment—n (%)	37 (62.7)	34 (57.6)	0.706

* Data available for 95 patients; § PESI was calculated for all patients a posteriori.

**Table 2 diagnostics-14-00776-t002:** Outcomes: percentage calculated on available data: * 25 patients; § 97 patients; ¶ 107 patients.

Outcomes	PESI (n = 59)	Routine Practice (n = 59)	*p*
LOS, days—median (IQR)	8 (6–12)	8 (6–12)	0.63
Discharge < 48 h—n (%)	0 (0.0)	0 (0.0)	
*In-hospital clinical outcomes*			
Death—n (%)	4 (6.8)	2 (3.4)	0.68
Recurrent VTE—n (%)	0 (0.0)	1 (1.7)	0.99
Major bleeding—n (%)	0 (0.0)	2 (3.4)	0.50
Heparin injection site hematoma *—n (%)	2 (14.3)	0 (0.0)	0.49
Heparin-induced thrombocytopenia—n (%)	0 (0.0)	0 (0.0)	
Hospital-acquired infections—n (%)	4 (6.8)	7 (11.9)	0.34
Iatrogenic complications—n (%) §	1 (2.1)	0 (0.0)	0.99
Immobilization syndrome—n (%) ¶	13 (24.5)	11 (20.4)	0.60
Pressure sores—n (%) ¶	5 (9.4)	7 (13.2)	0.54
*90-day clinical outcomes*			
Death—n (%)	5 (9.4)	3 (5.3)	0.48
Recurrent VTE—n (%)	0 (0.0)	2 (3.4)	0.50
Major bleeding—n (%)	0 (0.0)	2 (3.4)	0.50
New hospital admission—n (%)	4 (7.6)	4 (7.0)	0.99

LOS, length of hospital stay; h, hours; n, number; VTE, venous thromboembolism.

**Table 3 diagnostics-14-00776-t003:** Discharge destination: * chi-square *p* value relative to home discharge vs. other destinations.

	PESI (n = 59)	Routine Practice (n = 59)	*p*
Home—n (%)	48 (87.3)	49 (86.0)	0.84 *
Subacute/post-acute care facilities—n (%)	2 (3.6)	2 (3.5)	
Rehabilitation clinics—n (%)	0 (0.0)	0 (0.0)	
Nursing home—n (%)	5 (9.1)	6 (10.5)	

n, number.

**Table 4 diagnostics-14-00776-t004:** Clinical and family/social determinants of LOS: LOS, length of hospital stay; PE, pulmonary embolism; n, number; ys, years; RVD, right ventricular dysfunction.

	PESI (n = 59)	Routine Practice (n = 59)	*p*
Absence of caregiver—n (%)	10 (17.2)	10 (17.2)	0.97
Socio-familiar issues impacting discharge—n (%)	10 (17.2)	8 (14.3)	0.66
Incidental PE—n (%)	14 (23.7)	6 (10.2)	0.05
Hemodynamically unstable PE—n (%)	2 (3.4)	1 (1.7)	0.99
RVD—n (%)	17 (28.8)	22 (37.3)	0.33
Age, ys—mean (SD)	75.6 (13.1)	75.7 (12.7)	0.95
Active cancer—n (%)	12 (20.3)	11 (18.6)	0.82
Clinical complications—n (%)	14 (28.6)	18 (35.3)	0.47
Anemia (<12 g/dL)—n (%)	17 (29.8)	19 (32.2)	0.84
Thrombocytopenia (<100.000/mm^3^)—n (%)	2 (3.6)	2 (3.4)	0.96
Leucocytosis (>12.000/mm^3^)—n (%)	10 (18.2)	13 (22.0)	0.61

**Table 5 diagnostics-14-00776-t005:** Quality of life questionnaire: LOS, length of hospital stay; SD, standard deviation; PE pulmonary embolism.

Likert Scale	PESI (n = 59)	Routine Practice (n = 59)	*p*
Satisfied about hospitalization—mean (SD) (1, no; 5, very much)	3.8 (0.7)	3.8 (0.7)	0.89
Appropriate LOS (1, too low; 5, too high)	3.2 (0.7)	3.2 (0.5)	0.83
Worried about PE recurrence (1, no; 5, very much)	2.9 (1.0)	2.9 (0.8)	0.88
Worried about bleeding (1, no; 5, very much)	2.7 (1.0)	2.7 (0.8)	0.75

**Table 6 diagnostics-14-00776-t006:** Outcomes according to anticoagulant treatment regimen: DOAC, direct oral anticoagulant; LMWH, low-molecular-weight heparin; VKA, vitamin K antagonist; VTE, venous thromboembolism; h, hours.

Outcomes	DOAC (n = 71)	LMWH +/− VKA (n = 47)	*p*
LOS, days—median (IQR)	8 (5–11)	9 (7–12)	0.04
Discharge < 48 h—n (%)	0 (0.0)	0 (0.0)	
* In-hospital clinical outcomes *			
Death—n (%)	1 (1.4)	5 (10.6)	0.04
Recurrent VTE—n (%)	0 (0.0)	1 (2.1)	0.40
Major bleeding—n (%)	0 (0.0)	2 (4.3)	0.16
Heparin injection site hematoma—n (%)	0 (0.0)	2 (20.0)	0.15
Heparin-induced thrombocytopenia—n (%)	0 (0.0)	0 (0.0)	
Hospital-acquired infections—n (%)	7 (9.9)	4 (8.5)	0.99
Iatrogenic complications—n (%)	1 (1.8)	0 (0.0)	0.99
Immobilization syndrome—n (%)	11 (17.5)	13 (29.6)	0.14
Pressure sores—n (%)	2 (3.2)	10 (22.7)	0.003
* 90-day clinical outcomes *			
Death—n (%)	1 (1.6)	7 (14.9)	0.02
Recurrent VTE—n (%)	0 (0.0)	2 (4.3)	0.18
Major bleeding—n (%)	0 (0.0)	2 (4.3)	0.16
New hospital admission—n (%)	8 (12.7)	0 (0.0)	0.01

## Data Availability

The data are available upon request. The trial protocol is registered on ClinicalTrials.gov, identification number NCT03002467, and is fully accessible upon request to the corresponding author.

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
