# Peer review of "The Clinical Impact of the Pulmonary Embolism Severity Index on the Length of Hospital Stay of Patients with Pulmonary Embolism: A Randomized Controlled Trial"

_diagnostics, 2024, doi:10.3390/diagnostics14070776_

Round 1

Reviewer 1 Report

Comments and Suggestions for Authors

I am grateful to the editor for the opportunity to review the manuscript by Donadini et al, “Clinical impact of Pulmonary Embolism Severity Index on the length of hospital-stay in patients with pulmonary embolism: a randomized controlled trial.” In this article, the authors tried to study the hypothesis that the use of the Pulmonary Embolism Severity Index prediction scale can influence the reduction of length of hospital-stay in patients with pulmonary embolism. We must agree with the authors that this attempt was not successful. The scientific courage of the authors is respectable; it is usually not very common to report and publish negative research results.

However, while reviewing 3, I had comments to which I would like to receive responses from the authors.

1. In my opinion, the design of the study itself did not allow it to be completed (the study was completed ahead of schedule due to the slow enrollment of patients) and, accordingly, to obtain the expected results. It is obvious that the use of the well-known PESI prognostic scale in itself may not in any way affect the timing of treatment of patients. First of all, because in the control group no one prevents treating physicians from using it for the clinical assessment of patients. Accordingly, only any therapeutic interventions based on the PESI scale are able to influence the studied indicators. Which, in fact, was proven in a well-planned study, to which the authors also refer (ref. 28 in the article).

2. The insufficient number of patients included does not in itself allow any scientific conclusions to be drawn on this issue.

3. Table 1 does not contain statistical data comparing the studied groups, although they differed markedly in a number of indicators. Actually, the authors also admit this (“Indeed, there seem to be fewer severe PE cases (66% vs. 75%) and more patients with incidental PE (24% vs. 10%) in the PESI group compared to the standard of care group" - lines 289-291). Maybe these initial differences influenced the final result? This remains unclear.

4. Secondary analysis with comparison of VOCs in groups with different anticoagulant treatment regimens was carried out incorrectly, since there is no information about the initial condition of patients in the groups (it was necessary to provide information similar to Table 1). Therefore, it is unclear whether the anticoagulant regimen or the underlying milder clinical condition led to the differences in VOC.

5. I was surprised to discover that the article does not have a Results section; the information that should be placed in it belongs to the Methods section.

Comments on the Quality of English Language

No comments

Author Response

I am grateful to the editor for the opportunity to review the manuscript by Donadini et al, “Clinical impact of Pulmonary Embolism Severity Index on the length of hospital-stay in patients with pulmonary embolism: a randomized controlled trial.” In this article, the authors tried to study the hypothesis that the use of the Pulmonary Embolism Severity Index prediction scale can influence the reduction of length of hospital-stay in patients with pulmonary embolism. We must agree with the authors that this attempt was not successful. The scientific courage of the authors is respectable; it is usually not very common to report and publish negative research results.

We thank the reviewer for underlying the importance of publishing research results even if negative.

However, while reviewing 3, I had comments to which I would like to receive responses from the authors.

  1. In my opinion, the design of the study itself did not allow it to be completed (the study was completed ahead of schedule due to the slow enrollment of patients) and, accordingly, to obtain the expected results. It is obvious that the use of the well-known PESI prognostic scale in itself may not in any way affect the timing of treatment of patients. First of all, because in the control group no one prevents treating physicians from using it for the clinical assessment of patients. Accordingly, only any therapeutic interventions based on the PESI scale are able to influence the studied indicators. Which, in fact, was proven in a well-planned study, to which the authors also refer (ref. 28 in the article).

We thank the Reviewer for these valuable comments. Our aim was to provide evidence on the clinical impact of PESI, i.e. to demonstrate that adopting such a prognostic tool would have an impact on the outcome “length of hospital stay” (LOS).  Our hypothesis was based, at the time of projecting the study when more and more evidence was accumulating on the importance of stratifying PE prognosis, that the formal use of a validated tool such as PESI would have led to a better awareness and recognition of patients at low-risk of early adverse events, thus promoting a shorter hospital stay for them (please see the  Introduction section, page 2, line 66-71).

In addition, we agree with the Reviewer on the possible risk that the use of a similar strategy in clinical practice even for some patients belonging to the control group would have produced a reduction of LOS, thus preventing the trial from demonstrating our hypothesis. Therefore, we included this consideration for sample size calculation, as we anticipated that LOS would have been reduced by 5% in the standard of care arm. We added a further comment in the discussion section, please see it tracked on page 12, line 294-297 of the revised manuscript.

  1. The insufficient number of patients included does not in itself allow any scientific conclusions to be drawn on this issue.

We thank the reviewer for this comment. We acknowledged this limitation across the discussion section and added a further comment in the discussion section, on page 8, line 222-224 (please find it tracked in the revised manuscript)

  1. Table 1 does not contain statistical data comparing the studied groups, although they differed markedly in a number of indicators. Actually, the authors also admit this (“Indeed, there seem to be fewer severe PE cases (66% vs. 75%) and more patients with incidental PE (24% vs. 10%) in the PESI group compared to the standard of care group" - lines 289-291). Maybe these initial differences influenced the final result? This remains unclear.

We agree with the reviewer that some of the baseline characteristics of the patients may not be similar between the study groups. This is probably due to the lower-than-expected number of included patients, so that the randomization process was not able to distribute all the relevant characteristics equally between the study groups. Since this was not a pre-specified analysis because of the randomized design, we did not present a statistical test to compare those data. We now report here that the differences between the two study groups in terms of severe PE cases and incidental PE cases were not statistically significant (P value 0.4201 and P value 0.0859, respectively). We left to the editor the decision about including such results in Table 1.

  1. Secondary analysis with comparison of VOCs in groups with different anticoagulant treatment regimens was carried out incorrectly, since there is no information about the initial condition of patients in the groups (it was necessary to provide information similar to Table 1). Therefore, it is unclear whether the anticoagulant regimen or the underlying milder clinical condition led to the differences in VOC.

We thank the reviewer for this valuable comment. We did not include such a Table because it was part of a secondary analysis. Now we provide it here. We left to the editor the decision about including it in the main report or as supplementary data. We added some relevant results of that Table in the manuscript on page 7, line 206-211 (please find it tracked in the revised manuscript).

Patients, n

DOAC

 (n=71)

LMWH+/-VKA

(n=47)

P

Age, ys - mean (SD)

72.4 (13.4)

80.5 (10.2)

0.0006

Sex, Female - n (%)

31 (43.6)

23 (48.9)

0.706

Weight – mean (SD)

73.7 (13.8)

77.2 (12.1)

0.159

Unprovoked PE – n (%)

52 (73.2)

26 (55.3)

0.069

Active cancer – n (%)

8 (11.3)

15 (31.9)

0.011

Concomitant DVT – n (%)

42 (59.2)

23 (48.9)

0.366

Incidental PE – n (%)

10 (14.1)

10 (21.3)

0.442

SBP < 90 mmHg

1 (1.4)

2 (4.3)

0.562

RVD* on TTE– n (%)

13 (22.8)

13 (34.2)

0.330

RVD on Lab biomarkers

14 (19.7)

13 (27.7)

0.434

PESI – mean (SD)

105.5 (43.4)

129.9 (48.8)§

0.0053

PESI class III,IV,V – n (%)

43 (60.6)

40 (85.1)§

0.008

DVT deep vein thrombosis; DOAC direct oral anticoagulant; RVD right ventricular dysfunction; SBP: systolic blood pressure; TTE transthoracic echocardiography; ns not significant

*Data available on 95 patients (57 treated with DOAC, 38 treated with LMWH+/-VKA)

  • PESI was calculated for all patients a posteriori

  1. I was surprised to discover that the article does not have a Results section; the information that should be placed in it belongs to the Methods section.

We thank the reviewer for finding this formatting error. Indeed, the beginning of the Results section was erroneously left inside the last paragraph of the Methods section. We now fixed it, please find it tracked in the revised manuscript on page 4, line 158.

Reviewer 2 Report

Comments and Suggestions for Authors

The study and analysis were well done, but the hypothesis was hazardous since, besides the index, there are so many variables, as mentioned by the authors. I am not surprised that the patients' enrollment was slow. The only significant differences were related to the outcomes according to anticoagulant treatment, which may be biased by TE severity, mainly less severe TE treated with DOAC.

All the limitations are mentioned, but the clinical implications are equivocal.

Author Response

The study and analysis were well done, but the hypothesis was hazardous since, besides the index, there are so many variables, as mentioned by the authors. I am not surprised that the patients' enrollment was slow. The only significant differences were related to the outcomes according to anticoagulant treatment, which may be biased by TE severity, mainly less severe TE treated with DOAC.

We thank the Reviewer for these comments. We added some results regarding the comparison of patient groups according to the anticoagulant treatment, that support the hypothesis suggested by the Reviewer. Please see them tracked in the revised manuscript on page 7, line 206-211 and on the additional Table provided above in response to Reviewer#1 comment#4.

All the limitations are mentioned, but the clinical implications are equivocal.

We thank the reviewer for this comment. We added a further comment in the discussion section, on page 8, line 222-224 (please find it tracked in the revised manuscript)

Round 2

Reviewer 1 Report

Comments and Suggestions for Authors

The authors did some work to finalize the manuscript and responded to my comments. However, I am not entirely satisfied with these answers.

3. It does not matter what the original design of the study was. If authors compare 2 groups, data on statistical differences between them should be provided.

4. As I expected, the DOAC and LMWH+/-VKA groups were not comparable on baseline clinical parameters. Therefore, a table with these parameters must be provided. Also, taking into account such initial differences, the analysis presented by the authors in Table 6 is incorrect. That is, secondary analysis in this manuscript does not make sense.

5. The authors are still confused with the numbering of sections; for some reason, section 2. Results has the same numbering as section 2. Methods.

Comments on the Quality of English Language

No comments

Author Response

To the Editor in Chief of the Diagnostics,

We would like to thank you for giving us the opportunity to resubmit our manuscript entitled " Clinical impact of Pulmonary Embolism Severity Index on the length of hospital-stay in patients with pulmonary embolism: a randomized controlled trial ".

We appreciated the efforts made by the Referees for providing comments and suggestions, which, we believe, significantly improved the manuscript.

Kind Regards,

Nicola Mumoli and coauthors

Reply to Reviewers

Reviewer#1

The authors did some work to finalize the manuscript and responded to my comments. However, I am not entirely satisfied with these answers.

  1. It does not matter what the original design of the study was. If authors compare 2 groups, data on statistical differences between them should be provided.

We thank the Reviewer for this comment. Even if we do not fully agree with his/her statement about the need of applying statistical tests to compare the baseline characteristics of study groups of randomized trials, we provided p values for those comparisons in revised Table 1, none of which resulted statistically significant. Please see it tracked in Table 1 on page 5-6 of the revised manuscript.

  1. As I expected, the DOAC and LMWH+/-VKA groups were not comparable on baseline clinical parameters. Therefore, a table with these parameters must be provided. Also, taking into account such initial differences, the analysis presented by the authors in Table 6 is incorrect. That is, secondary analysis in this manuscript does not make sense.

As already included in the previous reply to reviewers’ comments, we did provide such a Table (copied also here). We also confirm here, as stated before, that we leave to the Editor the decision about including it in the main report or as supplementary data.

We regard to the secondary analysis presented in the paper, this was pre-specified in the protocol, based on the hypothesis that DOAC use could be associated with an easier management of anticoagulation in acute PE patients, thus potentially leading to a reduction of LOS. Indeed, based on this consideration, we also took into account such a variable in the randomization process, by stratifying it for the previously declared anticoagulant treatment choice of the local investigator, in order to avoid any influence of treatment choice on the final results (please see the Methods section on page3, line 140-144).

Therefore, it was expected that some difference could exist between patients treated with DOAC as compared to those LMWH+/-VKA, because they were not randomized to receive DOAC vs. LMWH+/-VKA. 

Based on all those considerations, that secondary analysis should not be considered “incorrect” and we remain convinced of the opportunity of presenting it, leaving to the Editor the final decision.

Table 7 (or Supplementary  Table).

Patients, n

DOAC

 (n=71)

LMWH+/-VKA

(n=47)

P

Age, ys - mean (SD)

72.4 (13.4)

80.5 (10.2)

0.0006

Sex, Female - n (%)

31 (43.6)

23 (48.9)

0.706

Weight – mean (SD)

73.7 (13.8)

77.2 (12.1)

0.159

Unprovoked PE – n (%)

52 (73.2)

26 (55.3)

0.069

Active cancer – n (%)

8 (11.3)

15 (31.9)

0.011

Concomitant DVT – n (%)

42 (59.2)

23 (48.9)

0.366

Incidental PE – n (%)

10 (14.1)

10 (21.3)

0.442

SBP < 90 mmHg

1 (1.4)

2 (4.3)

0.562

RVD* on TTE– n (%)

13 (22.8)

13 (34.2)

0.330

RVD on Lab biomarkers

14 (19.7)

13 (27.7)

0.434

PESI – mean (SD)

105.5 (43.4)

129.9 (48.8)§

0.0053

PESI class III,IV,V – n (%)

43 (60.6)

40 (85.1)§

0.008

DVT deep vein thrombosis; DOAC direct oral anticoagulant; RVD right ventricular dysfunction; SBP: systolic blood pressure; TTE transthoracic echocardiography; ns not significant

*Data available on 95 patients (57 treated with DOAC, 38 treated with LMWH+/-VKA)

  • PESI was calculated for all patients a posteriori

  1. The authors are still confused with the numbering of sections; for some reason, section 2. Results has the same numbering as section 2. Methods.

The numbering of Results section has been corrected, please see it tracked on page 4 of the revised manuscript.